# Apalutamide Monotherapy in Metastatic Hormone-Sensitive Prostate Cancer: A Viable Alternative to First-Generation Anti-Androgen Agents to Avoid the Flare Phenomenon and an Effective Treatment for Achieving Early PSA Response

**DOI:** 10.3390/cancers17152573

**Published:** 2025-08-05

**Authors:** Gaetano Facchini, Andrea D’Arienzo, Antonella Nicastro, Fabiano Flauto, Michela Izzo, Liliana Montella, Filippo Riccardo, Giovanni Maria Fusco, Francesco Trama, Giovanni Di Lauro, Giuseppe Di Costanzo, Anna Giacoma Tucci, Francesca Iasiello, Lorena Di Lorenzo, Salvatore Maddaluno, Carmela Liguori, Rita Assante di Cupillo, Paola Coppola, Angela Minissale, Maria Teresa Di Nardo, Luigi Formisano, Erika Martinelli, Giuliana Ciappina, Salvatore Pisconti, Massimiliano Berretta, Chiara Barraco

**Affiliations:** 1Oncology Complex Unit, Santa Maria delle Grazie Hospital, ASL NA2 NORD, 80078 Pozzuoli, Italy; gaetano.facchini@aslnapoli2nord.it (G.F.); andrea.darienzo@aslnapoli2nord.it (A.D.); antonella.nicastro@aslnapoli2nord.it (A.N.); michela.izzo@aslnapoli2nord.it (M.I.); liliana.montella@aslnapoli2nord.it (L.M.); carmela.liguori1@aslnapoli2nord.it (C.L.); oncologiapozzuoli.clinicaltrial@aslnapoli2nord.it (R.A.d.C.); paola.coppola@aslnapoli2nord.it (P.C.); angela.minissale@aslnapoli2nord.it (A.M.); mariateresa.dinardo@aslnapoli2nord.it (M.T.D.N.); 2Clinical Medicine and Surgery, University of Naples Federico II, 80131 Naples, Italy; fabiano.flauto@unina.it (F.F.); luigi.formisano1@unina.it (L.F.); 3Department of Precision Medicine, The University of Campania Luigi Vanvitelli, 80138 Naples, Italy; erika.martinelli@unicampania.it; 4Urology Complex Unit, Santa Maria delle Grazie Hospital, ASL NA2 NORD, 80078 Pozzuoli, Italy; filippo.riccardo@aslnapoli2nord.it (F.R.); giovannimaria.fusco@aslnapoli2nord.it (G.M.F.); francesco.trama@aslnapoli2nord.it (F.T.); giovanni.dilauro@aslnapoli2nord.it (G.D.L.); 5Radiology Complex Unit, Santa Maria delle Grazie Hospital, ASL NA2 NORD, 80078 Pozzuoli, Italy; giuseppe.dicostanzo@aslnapoli2nord.it (G.D.C.); anna.tucci@aslnapoli2nord.it (A.G.T.); francesca.iasiello@aslnapoli2nord.it (F.I.); 6Clinical Pathology Unit, Santa Maria delle Grazie Hospital, ASL NA2 NORD, 80078 Pozzuoli, Italy; lorena.dilorenzo@aslnapoli2nord.it (L.D.L.); salvatore.maddaluno@aslnapoli2nord.it (S.M.); 7Department of Medical Sciences, Section of Experimental Medicine, University of Ferrara, 44121 Ferrara, Italy; giuliana.ciappina@unife.it; 8Oncology Unit, Ospedale San Giuseppe Moscati, Statte, 74010 Taranto, Italy; salvatore.pisconti@unita.it; 9Division of Oncology, Azienda Ospedaliera Universitaria “Gaetano Martino” Hospital, 98124 Messina, Italy; massimiliano.berretta@unime.it; 10Department of Clinical and Experimental Medicine, University of Messina, 98122 Messina, Italy

**Keywords:** apalutamide, flare phenomenon, metastatic hormone-sensitive prostate cancer, treatment, real-world analyses, androgen deprivation therapy, androgen receptor targeted agents

## Abstract

In this manuscript, we describe our clinical experience with patients diagnosed with metastatic hormone-sensitive prostate cancer (mHSPC) treated in first-line with apalutamide monotherapy for 14 days (to prevent the flare-up phenomenon) followed by combination therapy with apalutamide plus a gonadotropin-releasing hormone (GnRH) agonist. This treatment was continued until progression of the disease or unacceptable toxicities. We collected and analyzed serum PSA and testosterone levels at three time points: baseline (prior to starting the apalutamide monotherapy), day 14 (after 13 days of apalutamide alone), and day 28 (after 15 days of combination therapy with apalutamide and GnRH agonist), in order to assess the depth and speed of the biochemical response. Additionally, we evaluated adverse events during treatment and monitored the achievement of castration status.

## 1. Introduction

Prostate cancer (PCa) is the second most commonly diagnosed cancer in men, with an estimated 1.4 million diagnoses and 375,000 deaths worldwide in 2020 [1,2]. In more than half of the countries around the world, it is the most frequently diagnosed cancer in men and the leading cause of death among men in a quarter of all countries. In Europe, it is the most frequently diagnosed cancer in men and the third most common cancer-related cause of death in men [3]. Incidence of PCa diagnosis varies between different geographical areas. It is highest in Australia/New Zealand and Northern America (with age-standardized rates [ASR] per 100,000 of 111.6 and 97.2, respectively), and in Western and Northern Europe (ASRs of 94.9 and 85, respectively) [4], while it is lower but rising in Eastern and South-Central Asia (ASRs of 10.5 and 4.5, respectively) [5]. A steady increase in PCa rates has been found in Eastern and Southern Europe [2,6]. Among the recognized risk factors are hereditary (ethnicity, family history and known genetic mutations) and non-hereditary types, such as dietary factors, as well as metabolic syndrome and obesity. Even though germline mutations leading to PCa are relatively rare (1/300), the impact on PCa risk is quite strong. Pathogenic variants were most commonly identified in BRCA2 (4.5%), CHEK2 (2.2%), ATM (1.8%), and BRCA1 (1.1%) [7].

The diagnostic pathway starts as follows: evaluation of clinical symptoms, prostate-specific antigen (PSA) detection, and digital rectal examination (DRE). Symptoms usually occur late in the natural history of PCa and localised PCa is therefore usually asymptomatic. Local progression may cause symptoms such as lower urinary tract symptoms (LUTS), erectile dysfunction (ED), retention, pain, haematospermia or haematuria. Bone metastases may cause pain or spinal cord compression. PSA is a glycoprotein enzyme secreted by prostate epithelial cells with a small portion present in the blood stream. It is organ specific but not cancer specific; therefore, it may also be elevated in benign prostatic hyperplasia (BPH), prostatitis and other non-malignant conditions [8]. Magnetic resonance imaging (MRI) represents the first diagnostic tool after suspicion of PCa and provides staging information, also allowing for guidance in targeted prostate biopsy. The Prostate Imaging-Reporting and Data System (PI-RADS) standardizes interpretation and stratifies men with suspected PCa on a one-to-five risk scale of having PCa [9]. MRI and transrectal ultrasound (TRUS) could complement each other [4]. Diagnosis of PCa is based on histology; targeted transperineal biopsies, in comparison with systematic transrectal biopsies, result in an increased detection rate in clinically significant prostate cancer. Each biopsy should be reported individually and evaluated using the International Society of Urological Pathology Consensus recommendations [10]. ISUP GG (Global International Society of Urological Pathology Grade Group) figures among the mandatory elements provided by the pathology report, representing the strongest prognostic factor for clinical behavior and treatment response [11]. Regarding staging assessment, magnetic resonance imaging provides T staging and can inform surgical technique. For patients with very-low-, low-, and intermediate-risk prostate cancer and a life expectancy of five years or less and without clinical symptoms, further imaging and treatment should be delayed until symptoms develop, at which time imaging can be performed and ADT should be given. Those with a life expectancy less than or equal to five years who fall into the very-high-risk categories should undergo bone imaging and, if indicated, pelvic +/− abdominal imaging. For symptomatic patients and/or those with a life expectancy of greater than five years, bone and soft tissue imaging is appropriate for patients with unfavorable intermediate-risk, high-risk, and very-high-risk prostate cancer. Bone imaging can be achieved using conventional technetium-99mMDP bone scans while CT, MRI, or PET/CT or PET/MRI can be considered for equivocal results on initial bone imaging. Soft tissue imaging of the pelvis, abdomen, and chest can include chest CT and abdominal/pelvic CT or abdominal/pelvic MRI (preferred over CT for pelvic staging). Alternatively, Ga-68 PSMA-11 or F-18 piflufolastat PSMA PET/CT or PET/MRI can be considered for bone and soft tissue (full body) imaging, considering its increased sensitivity and specificity for detecting micrometastatic disease compared to conventional imaging. PSMA-PET refers to a growing body of radiopharmaceuticals that target prostate specific membrane antigen (PSMA) on the surface of prostate cells. Currently, there are five PET tracers that are FDA approved for use in patients with prostate cancer: Ga-68 PSMA-11 (PSMA-HBED-CC), F-18 piflufolastat (DCFPyL), C-11 choline, F-18 fluciclovine, and F-18 sodium fluoride. Although these tracers are approved for the evaluation of patients with biochemical recurrence, the PSMA tracers Ga-68 PSMA-11 and F-18 piflufolastat are also approved for patients at initial staging with suspected metastatic disease. Tracer distribution in patients with prostate cancer can be imaged with either PET/CT or PET/MRI modalities. Although CT and MRI are equivalent in the assessment of lymphadenopathy, PET/MRI has the added advantage over PET/CT with enhanced tissue contrast, which is especially important in evaluation of pelvic anatomy and prostate cancer assessment [12]. The evaluation of life expectancy and overall health status is essential in guiding clinical decision making for the early detection, diagnosis, and treatment of prostate cancer (Pca). Clinical decisions should be based not only on chronological age but also on a comprehensive evaluation of individual life expectancy, functional status, frailty, and comorbidities.

Androgen deprivation therapy (ADT) is the mainstay of prostate cancer treatment, especially in metastatic disease. While it can be omitted in low-risk localized PCa according to D’Amico classification risk class [7,12], patients with locally advanced or metastatic PCa undergo life-long ADT. At present, androgen deprivation is achieved through pharmaceutical interventions; surgical bilateral orchiectomy, employed for the first time by Huggins and Hodges in 1941 to improve outcomes in men with advanced prostate cancer, is no longer considered an option [13]. The main androgen hormone is testosterone, which plays an important role both in prostate physiology and pathology. Therefore, androgen deprivation consists of reducing serum testosterone levels, also referred to as castration, to < 50 ng/dL. About 90% of testosterone is produced by Leydig cells in the testis under Luteinizing Hormone (LH) stimulation. The pituitary gland centrally governs this process through gonadotropin-releasing hormone (GnRH) secretion from its anterior lobe, which regulates LH and follicle-stimulating hormone (FSH) production and, therefore, testosterone levels [14]. Pharmacological castration involves GnRH analogue drugs, which bind GnRH receptors in the pituitary gland with the final consequence of inhibiting testosterone production. GnRH analogues can be agonists (such as Leuprorelin, Goserelin, and Triptorelin) administered as an intramuscular or subcutaneous injection, or antagonists (such as Degarelix or Relugolix), administered orally or as a subcutaneous injection [13]. Notably, the non-pulsatile stimulation of GnRH receptors by agonists initially causes a surge in LH and testosterone, defined as a “testosterone surge” or “flare-up phenomenon”; however, sustained continuous stimulation results in receptor downregulation over a few weeks [15]. The flare-up phenomenon is clinically significant in about 10% of patients treated with GnRH agonists but is not associated with antagonists, which cause rapid reductions in serum testosterone without the initial surge [16]. Testosterone levels increase by a maximum of two-fold starting on days two to four and decrease to baseline values in about one week [17]. The correlation between testosterone flare and the progression of prostate cancer has not been proven, but this phenomenon could in theory increase bone pain, spinal cord compression, bladder outlet obstruction and cardiovascular problems [18]. However, while GnRH agonist monotherapy seems to be safe in patients with low-stage disease and low bone involvement, the testosterone surge could exacerbate complications in patients with advanced disease, i.e., high-volume bony disease [7,18].

Due to its lipophilic structure, testosterone easily enters into the cell and binds the androgen receptor (AR) at the ligand binding domain, causing dimerization and translocation of AR to the nucleus with subsequent activation or repression of its target genes together with co-regulators. AR target genes can influence a number of cancer-relevant cellular processes, such as cell cycle, cell death, metabolism, chromatin remodeling, invasion and DNA repair, playing a critical role in cancer metabolism, proliferation, survival and invasion [14]. As a result of the indispensable role of AR in prostate cancer, a number of anti-AR drugs have been developed and approved for different stages of prostate cancer in the past 30 years. The first-generation AR antagonists included flutamide, nilutamide, and bicalutamide, which were approved by the FDA in 1989, 1995, and 1996, respectively [19]; they were extensively employed in the past for PCa management, both alone or combined with GnRH analogues. While the patients respond to first-generation AR antagonists in the early stages of the disease, they eventually acquire resistance and progress to lethal-stage castration-resistant prostate cancer (CRPC) [20]. A growing dataset indicates that the restoration of AR signaling is critical for disease progression in these patients, as AR overexpression, especially due to AR genomic amplification, has been frequently observed and proven to be a principal driver of prostate cancer progression, both in clinical CRPC patients and in preclinical prostate cancer cell models [21]. The continued importance of the AR pathway in CRPC has encouraged researchers and clinicians to develop a second generation of AR antagonists with higher AR binding affinity and specificity to target aberrant AR signaling in lethal-stage CRPC patients. At present, considering the above-mentioned rationale, the therapeutic application of first-generation antagonists has been significantly downscaled. Indeed, a short course of bicalutamide (three to four weeks) is generally administered at the beginning of a GnRH agonist treatment in order to control the possible initial increase in testosterone, eventually impacting disease progression. The European Association of Urology (EAU) and the National Comprehensive Cancer Network (NCCN) guidelines recommend the association of a first-generation anti-androgen with GnRH agonists to prevent the flare-up phenomenon in patients with high-volume bony disease; the recommendation suggests administering the anti-androgen for at least seven days before starting GnRH in order to manage potential increases in testosterone levels and mitigate risks of adverse events (weak recommendation) [12,16,18].

The current hormonal treatment strategies for advanced prostate cancer are based on two principal therapeutic approaches: inhibition of conversion of extragonadal precursor steroids to testosterone and 5α-dihydrotestosterone (DHT) with abiraterone and direct blockade of the AR to prevent binding to its ligands, testosterone, and DHT with next-generation AR antagonists, such as apalutamide, darolutamide, or enzalutamide [22]. Abiraterone acetate is an oral prodrug that is deacetylated to its active metabolite abiraterone, which inhibits the steroidogenic CYP17A1 enzyme, a member of the cytochrome P450 family. Abiraterone is currently indicated in the United States for the treatment of mCRPC and metastatic castration-sensitive prostate cancer (mCSPC), in combination with prednisone, according to the results of several trials, such as COU-AA-301 [23], COU-AA-302 [24], LATITUDE [25] and STAMPEDE [26]. Second-generation androgen receptor targeted agents (ARTA), i.e., Apalutamide, Enzalutamide and Darolutamide, exhibit more efficient inhibition of the AR pathway compared to first-generation ones: not only do they bind to the ligand binding domain, but they also prevent AR translocation into the nucleus and the recruitment of co-activators [19]. ARTA demonstrated a significant improvement in oncological outcomes in hormone-sensitive and castration-resistant prostate cancer, either metastatic (mHSPC and mCRPC) or high-risk non-metastatic (nmHSPC and nmCRPC) [27].

Enzalutamide is indicated for the treatment of both CRPC and mHSPC, either as a first-line or second-line therapy, depending on the clinical context. It was initially approved in the post-docetaxel setting based on the pivotal AFFIRM trial [28]. Subsequently, the PREVAIL trial demonstrated the superior efficacy of enzalutamide compared to placebo in patients with CRPC who had not previously received chemotherapy [29]. Following the clinical success of enzalutamide in the docetaxel-naïve setting, investigators completed the PROSPER trial to demonstrate its efficacy in delaying metastases in patients with high-risk (PSA doubling time <10 months) non-metastatic CRPC (nmCRPC) [30]. The success of the ENZAMET [31] and ARCHES [32] trials led to the expansion of enzalutamide’s clinical indications to include mHSPC. Moreover, it was recently approved for the treatment of biochemically recurrent prostate cancer (BCR), following the results of the EMBARK trial [33].

Darolutamide was approved for the treatment of high-risk non-metastatic castration-resistant prostate cancer (nmCRPC) in combination with ADT, based on the results of the ARAMIS trial [34]. Furthermore, the ARASENS trial demonstrated that, in patients with high-volume and high-risk or low-risk metastatic castration-sensitive prostate cancer (mCSPC), treatment intensification with darolutamide, ADT, and docetaxel improved overall survival (OS) [35], leading to its approval in this setting. More recently, the results of the ARANOTE trial, which confirmed the efficacy and tolerability of darolutamide plus ADT in patients with mHSPC [36], led to the expansion of darolutamide’s clinical indications.

Apalutamide is a selective, nonsteroidal, competitive AR inhibitor that was the result of a discovery effort using structure–activity relationship chemistry to find molecules with full AR antagonist activity. This small-molecule drug binds to the AR ligand-binding domain, inhibiting AR translocation into the nucleus, DNA binding, and downstream transcription of AR-related genes. Apalutamide binds AR with 7- to 10-fold greater affinity than bicalutamide. The antitumor activity of bicalutamide was largely restricted to growth inhibition rather than tumor shrinkage, while apalutamide induced significant tumor regression [37]. Currently approved indications in the United States include mHSPC and nmCRPC. In the TITAN clinical trial, the combination of apalutamide and ADT, compared to ADT plus placebo, demonstrated in 525 patients with mHSPC a reduction of 52% for the risk of radiological progression [38] and 48% for the risk of death after adjusting for crossover [39]. Overall survival rates at two years in the clinical trial and in real-world analysis exceed 80% (even 90% in several subgroups) [40] and median overall survival is still not reached; the statistical estimation of median overall survival in patients with mHSPC is about 72 months, with 110 months in low-volume and 52 months in high-volume disease [41]. Moreover, Chowdhury et al. recently showed that PSA level reduction has a prognostic role in the TITAN trial. Apalutamide lead to a ≥50% PSA levels reduction in 90% vs. 55% of patients in the placebo group. Furthermore, the achievement of ≥90% PSA reduction or PSA values ≤0.2 ng/mL (defined as undetectable) at three months was associated with improvement in overall survival, radiological progression-free survival, time to PSA progression and time to castration resistance, compared with patients who did not achieve a decline in PSA [42]. Consequently, PSA measurement represents a well-established tool not only for screening and diagnosis, but also for monitoring therapeutic response in PCa. Indeed, early and deep PSA response has been associated with improved prognosis in metastatic Pca; it has also been associated with, or showed surrogacy for, better clinical outcomes, including overall survival.

The SPARTAN trial enrolled patients with high-risk nmCRPC into an apalutamide treatment arm or placebo arm, demonstrating a significantly longer (>two years) median metastasis-free survival (MFS) with apalutamide compared to placebo. This corresponded to a 70% reduction in the risk of death or metastasis with apalutamide treatment, as well as statistically significant improvement in all secondary endpoints, including progression-free survival (PFS), time to metastasis, and time to symptomatic progression [43]. These results led to its approval in high-risk nmCRPC. Even in this trial, Apalutamide reduced PSA levels in about 90% of patients (compared to 2% of placebo group) and a ≥90% PSA reduction or undetectable levels were significantly associated with improved outcomes [44].

In the TITAN and SPARTAN trials, treatment with first-generation antiandrogen for 14 or more days was required before randomization and so before apalutamide administration. This approach accounts for the double-blind design of the trials and addresses the flare phenomenon in the placebo arm [35,38].

Overall, given the above-mentioned considerations, ADT and ARTA are associated with improved survival in PCa treatment. However, they are also associated with significant adverse events. In addition to prostate physiology and pathology, testosterone plays an important role in other organs such as bone, muscle, and the circulatory system. Androgen deprivation is associated with decrease in bone mineral density and increases the risk of fractures proportionally to ADT duration. Therefore, bone-targeted agents should be administered during ADT according to international guidelines [7,12,45]. Testosterone decrease can alter body composition, with reduced muscular mass and an increased body fat percentage, resulting in weight gain and alterations in insulin sensitivity. As a consequence, ADT is associated with metabolic syndrome and increased cardiovascular risk and mortality. Other adverse events that significantly impact quality of life are loss of libido, erectile dysfunction, fatigue, hot flushes, and cognitive decline, as well as the relationship with the patient’s partner [45]. Adverse events associated with ARTA in clinical trials mostly overlap with ADT because they are used in combination; however, they are associated with increased risk of hypertension, hot flushes, headaches, and falls [46]. Although generally well tolerated, Apalutamide is associated with specific adverse events such as skin rash (27%), pruritus (10%) and hypothyroidism (6.5%) [38]. Moreover, it is associated with a low rate of serum enzyme elevation during therapy but has not been linked to cases of clinically apparent liver injury with jaundice. Unlike apalutamide, although rarely, first-generation anti-androgens (mainly Flutamide and Bicalutamide) could be associated with fulminant hepatotoxicity [47].

Considering the more efficient block of AR by ARTA, the use of these drugs following a first-generation agent determine only a redundancy in pharmacological action without a proven benefit. Furthermore, given the prognostic role of a rapid and deep PSA decline in patients treated with apalutamide on oncological outcome, it is of paramount importance to start the treatment with the most effective drug to accelerate the PSA response. In the present report, we question the real utility of the association of first-generation antiandrogen to GnRH agonists at the beginning of an ARTA treatment. In fact, we describe our clinical experience about patients diagnosed with mHSPC who received first-line apalutamide monotherapy for 14 days (to prevent flare-up phenomenon) followed by combination therapy with apalutamide plus GnRH agonist. This treatment was continued until progression disease or unacceptable toxicities.

Although for a short period, our trial explores ARTA monotherapy in metastatic PCa. In this study, differently from the mentioned ARTA clinical trials, apalutamide at the dose of 240 mg was administered for an induction period of 14 days and subsequently GnRH agonists were administered in combination every 28 days. Monotherapy with Apalutamide, Enzalutamide and Darolutamide was evaluated in several clinical trials in the setting of high-risk biochemical recurrence: to date, as mentioned above, only enzalutamide is approved in this setting because of survival benefit in these patients compared to ADT alone, although it was inferior to Enzalutamide and ADT combination. As a consequence of the mechanism of action, ARTA monotherapy is associated with low incidence of hot flushes but higher incidence of breast-related adverse events as gynecomastia and breast pain: low AR stimulation causes an increase in circulating testosterone levels, which are converted to estrogen through the aromatase enzyme and cause breast-related toxicity [48]. Therefore, in our study, we also aim to assess any potential adverse events related to Apalutamide monotherapy during the first 14 days of treatment, as well as during the subsequent combination therapy with apalutamide plus GnRHagonist in real-word settings.

## 2. Materials and Methods

This is a descriptive retrospective study of 27 patients with mHSPC who received first-line treatment with apalutamide monotherapy for 14 days (to prevent the flare-up phenomenon) followed by continuous combination with a GnRH agonist plus apalutamide at the Oncology Complex Unit of the “S. Maria delle Grazie” Hospital, Pozzuoli, Italy, between June 2022 and April 2024. Inclusion criteria were a confirmed diagnosis of mHSPC, absence of previous hormonal treatments for metastatic disease, and provided informed consent to participate. Patients included in other clinical trials were excluded. The protocol was approved by the Campania 1 Ethical Committee.

Treatment consisted of apalutamide monotherapy at a dose of 240 mg orally once per day from day 1 to day 13, subsequently associated with a GnRH agonist (Triptorelin or Leuprorelin) from day 14. The administration of GnRH agonists was repeated every 28 days. Treatment combination was continued until progression or unacceptable toxicity occurred, according to clinical practice. Serum PSA and testosterone levels were measured at baseline (at starting frontline treatment with apalutamide alone), at day 14 (after 13 days of apalutamide monotherapy), at day 28 (after additional 15 days of apalutamide plus a GnRH agonist), and at day 60. Undetectable PSA was defined as a value ≤0.2 ng/mL. PSA and testosterone were dosed with blood samples and tested either in our hospital or in external laboratories with a chemiluminescence assay.

Selected patients and disease characteristics were recorded at diagnosis and/or at starting frontline apalutamide treatment, including age, Gleason score, tumor stage, presence of comorbidities, Eastern Cooperative Oncology Group performance status (ECOG PS) and disease volume, classified according to the definition given in the CHAARTED trial [49].

### Statistical Analysis

Descriptive analyses were conducted to characterize PSA and testosterone changes over time, as well as safety. Normally distributed continuous variables are presented as mean values ± standard deviations, and categorical variables are reported as numbers and percentages. Comparisons between groups (i.e., low- vs. high-volume disease) were performed using the contingency table analysis with the Chi-square or Fisher’s exact test, as appropriate (for categorical data), and through a Wilcoxon rank–sum test (for continuous data). Adverse events were tabulated, both overall and according to their grade. Analyses were conducted using SAS version 9.4 (SAS Institute, Cary, NC, USA).

## 3. Results

### 3.1. Baseline Patients’ Characteristics

This study focused on a total of 27 mHSPC patients treated with apalutamide monotherapy, which was followed, after 14 days, by the addition of a GnRH agonist. At prostate cancer diagnosis, the mean patient age was 68.2 (±6.9) years, with 15 patients (44.4%) presenting de novo metastases. At the start of the frontline apalutamide treatment, the mean age was 71.1 (±7.4) years. The majority of patients (85.2%) had ECOG PS of 0, and 17 (63.0%) and 10 (37.0%) patients had, respectively, low- and high-volume disease. The Gleason score was equal to 7 in 8 patients (30.8%), 8 in 12 patients (46.1%) and ≥9 in 6 patients (23.1%) (the sum does not add up to the total because of 1 missing data point—see Table 1).

### 3.2. Decrease in PSA Levels

PSA levels decreased from a mean of 45.2 (±63.1) ng/mL at baseline to a mean of 12.6 (±23.4) ng/mL at day 14 and to 3.3 ng/mL (±6.0) at day 28 (Table 2). After 14 days of apalutamide monotherapy, 21 patients (77.8%) achieved a >50% PSA reduction and 4 (14.8%) a >90% PSA reduction (Figure 1). At day 28 (after an additional 14 days of apalutamide monotherapy and 14 days of apalutamide plus GnRH agonist), 27 patients (100%) achieved a >50% PSA reduction and 13 (48.1%) a >90% PSA reduction. The number of patients with undetectable PSA was 1 (3.7%) at day 14, 2 (7.4%) at day 28 and 9 (33.3%) at day 60. Overall, 20 patients (74.1%) achieved undetectable PSA at any time in the first 28 days of treatment. No significant difference in PSA levels or PSA reduction emerged between subgroups of patients with low- vs. high-volume disease (all *p*-values were >0.05).

### 3.3. Decrease in Testosterone Levels

The mean serum testosterone levels were 6.56 (±4.46) ng/mL at baseline, 6.58 (±4.42) ng/mL at day 14, and 2.40 (± 3.38) ng/mL at day 28 (Appendix A). A total of six patients (22.2%) had a reduction in testosterone levels at day 14, with one patient experiencing a >90% reduction from the baseline value. At day 28, 24 patients (88.9%) showed a testosterone decrease, with 20 patients (74.1%) achieving a >50% and 8 (29.6%) a >90% reduction. No significant difference in testosterone levels or testosterone decrease was reported across subgroups of disease volume (all *p*-values were >0.05).

### 3.4. Safety

During treatment, 16 patients (59.3%) reported adverse events, the most common being those of the skin (n = 11, 40.7%, [G1–2: n = 6, G3: n = 5]). Three patients (11.1%) had hypertension, one (3.7%) reported cardiopathy and one (3.7%) hypothyroidism. No patients reported signs or symptoms caused by the flare-up phenomenon during the observation period (see Appendix A).

## 4. Discussion

This is one of the first studies describing, as a single-center experience, the biochemical response and safety of frontline apalutamide monotherapy followed by GnRH agonist administration in a series of mHSPC patients without the need for first-generation anti-androgen to avoid the flare-up phenomenon. We observed a rapid lowering in PSA levels after two weeks of apalutamide monotherapy treatment, which further declined after an additional two weeks of the combination apalutamide + GnRH agonist. Furthermore, all patients reached optimal castration levels in the first month of treatment. No new adverse events were reported, and the safety profile was consistent with previous studies.

The efficacy of the apalutamide plus GnRH agonist therapy on PSA levels, as well as the association between deep and rapid PSA control and clinical outcomes, was demonstrated for patients with mHSPC in the TITAN phase III trial [38]. In our cohort, over two-thirds of patients had >50% PSA level reduction (77.8%), with 15% achieving a >90% decline, as soon as only 14 days of treatment with apalutamide monotherapy, thus demonstrating the high AR-binding affinity and efficacy of the drug, even in the normal-testosterone-level phase in the absence of combined GnRH agonist therapy. Importantly, the effect of apalutamide on the decrease in PSA values was rapid, with noticeable results apparent after just 14 days of treatment. Literature data on this topic from real-world studies are particularly scanty. Our findings thus provide useful new insights of potential relevance for the management of mHSPC in everyday clinical practice.

Various international guidelines recommend short-term use of anti-androgenic treatment in newly diagnosed HSPC cases before starting GnRH agonists in order to mitigate the consequences of flare-up phenomena, although the evidence is weak [7,12]. Our results support the use of apalutamide alone to protect against testosterone surge, which allows for PSA control, avoiding first-generation anti-androgens (bicalutamide, flutamide, nilutamide). The mean testosterone level, in fact, remained materially unchanged after two weeks of apalutamide alone in the majority of patients, while it decreased strongly (by almost two-thirds) after two subsequent weeks of apalutamide + GnRH agonist therapy. Previous information is limited, particularly from the clinical practice setting. In a case report from China, one patient with newly diagnosed mHSPC underwent the same treatment schedule, reporting an upward trend in testosterone levels during the first two weeks, followed by a rapid decline during the period of combined therapy with apalutamide and GnRH agonists [50]. Similar results emerged in a phase II open-label trial including a study arm of 42 patients with advanced HSPC treated with apalutamide alone [48,49].

The major limitations of this study are its observational design and mainly descriptive aims. No comparison groups were available and the size of the mHSPC population was limited. Still, this exploratory analysis from a real-life setting may provide relevant clinical indications and promote new studies on the topic.

## 5. Conclusions

According to our findings, apalutamide alone is a viable option for mitigating the flare-up phenomenon while avoiding first generation anti-androgen therapy (bicalutamide, flutamide, nilutamide); it is able to achieve a rapid and deep biochemical control and allows for the most effective therapy to be started earlier. In conclusion, our findings support a significant change in clinical practice for first-line patients with metastatic hormone-sensitive prostate cancer (mHSPC) who can benefit immediately from a new androgen-receptor-targeting agent (ARTA) such as apalutamide to mitigate the flare-up phenomenon and to achieve a rapid and deep biochemical control. Further studies are needed to confirm this result.

## Figures and Tables

**Figure 1 cancers-17-02573-f001:**
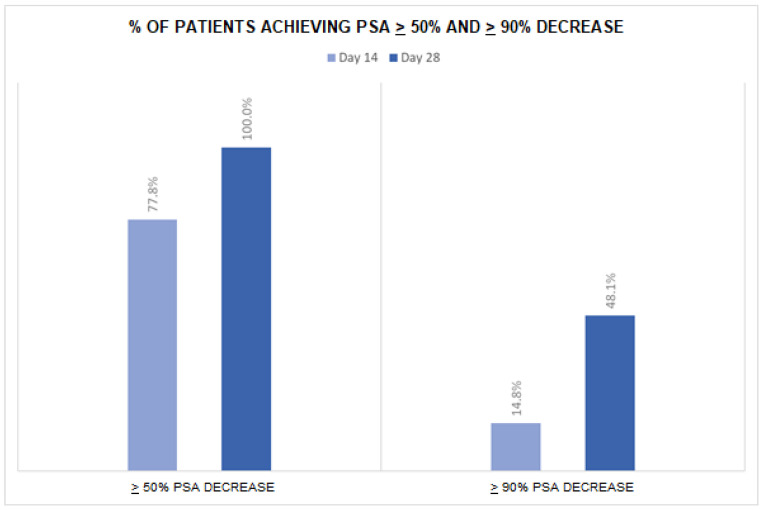
Proportion of patients achieving selected targets of PSA decrease during the follow-up period.

**Table 1 cancers-17-02573-t001:** Main baseline characteristics of 27 mHSPC patients at the start of apalutamide monotherapy (14-day period) for first-line treatment.

	All Patients (n = 27) n (%)
Age at cancer diagnosis (years)	
<65	9 (33.3)
65–74	13 (48.2)
≥75	5 (18.5)
Mean age (SD)	68.2 (6.9)
Stage at cancer diagnosis	
M0	12 (44.4)
M1	15 (55.6)
Gleason score (at cancer diagnosis) ^a^	
7	8 (30.8)
8	12 (46.1)
≥9	6 (23.1)
Age at starting first line treatment (years)	
<65	5 (18.5)
65–74	13 (48.2)
≥75	9 (33.3)
Mean age (SD)	71.1 (7.4)
Performance Status (at starting first line treatment)	
0	23 (85.2)
1	4 (14.8)
Disease volume (CHAARTED criteria)	
Low-volume disease	17 (63.0)
High-volume disease	10 (37.0)

mHSPC: metastatic hormone-sensitive prostate cancer; SD: standard deviation. ^a^ The sum does not add up to the total because of 1 missing information.

**Table 2 cancers-17-02573-t002:** PSA levels and proportion of patients achieving selected PSA targets at 14 and 28 days, overall and according to disease volume.

	All Patients (n = 27)	Low Volume (n = 17)	High Volume (n = 10)	*p*-Value ^c^
PSA at baseline, Mean (SD)	45.2 (63.1)	33.4 (43.6)	65.3 (86.1)	0.38
PSA at day 14, Mean (SD)	12.6 (23.4)	11.7 (23.5)	14.2 (24.4)	0.50
PSA at day 28, Mean (SD)	3.3 (6.0)	2.9 (6.0)	3.9 (6.4)	0.37
Achieved a ≥50% PSA decrease at day 14 ^a^	n (%)			
No	6 (22.2)	5 (29.4)	1 (10.0)	
Yes	21 (77.8)	12 (70.6)	9 (90.0)	0.36
Achieved a ≥90% PSA decrease at day 14 ^a^				
No	23 (85.2)	14 (82.3)	9 (90.0)	
Yes	4 (14.8)	3 (17.7)	1 (10.0)	0.99
Achieved a ≥50% PSA decrease at day 28 ^a^				
No	0 (0.0)	0 (0.0)	0 (0.0)	
Yes	27 (100.0)	17 (100.0)	10 (100.0)	-
Achieved a ≥90% PSA decrease at day 28 ^a^				
No	14 (51.8)	10 (58.8)	4 (40.0)	
Yes	13 (48.1)	7 (41.2)	6 (60.0)	0.44
Patients with undetectable PSA at day 14 ^b^	1 (3.7)	1 (5.9)	0 (0.0)	0.99
Patients with undetectable PSA at day 28 ^b^	2 (7.4)	2 (11.8)	0 (0.0)	0.52
Patients with undetectable PSA at day 60 ^b^	9 (33.3)	6 (35.3)	3 (30.0)	0.99
Patients with undetectable PSA (any time during follow-up) ^b^	20 (74.1)	13 (76.5)	7 (70.0)	0.99

^a^ As compared to day 0. ^b^ Undetectable PSA defined as PSA < 0.02 ng/mL. ^c^
*p*-value for comparison between low- and high-volume disease. PSA: prostate-specific antigen; SD: standard deviation.

## Data Availability

The data presented in this study are available on request from the corresponding author due to privacy.

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
