# Peer review of "Apalutamide Monotherapy in Metastatic Hormone-Sensitive Prostate Cancer: A Viable Alternative to First-Generation Anti-Androgen Agents to Avoid the Flare Phenomenon and an Effective Treatment for Achieving Early PSA Response"

_cancers, 2025, doi:10.3390/cancers17152573_

Round 1

Reviewer 1 Report

Comments and Suggestions for Authors

This is a nice, albeit an unoriginal study. Although personally, I like real-world evidence, with all the noise of day-to-day clinical practice. 

One of the major limitations of the study is its duration. The clinical study with apalutamide lasted 54 months (TITAN) or over 40 months (SPARTAN). Each of the two studies included more than 1,000 patients (source: SmPC). So the question arises: what does this study bring what has not been already shown by clinical trials?

Minor thing: authors refer to Toxicity; ususal it is referred to as Adverse reactions. Toxicity is usually related to severe/extreme adverse reactions with clinical toxicity.

Author Response

Comment 1: [This is a nice, albeit an unoriginal study. Although personally, I like real-world evidence, with all the noise of day-to-day clinical practice. One of the major limitations of the study is its duration. The clinical study with apalutamide lasted 54 months (TITAN) or over 40 months (SPARTAN). Each of the two studies included more than 1,000 patients (source: SmPC). So the question arises: what does this study bring what has not been already shown by clinical trials?]

Response 1: Thank you for pointing this out. The main goal of our study is to evaluate the use of Apalutamide monotherapy for first 14 day in order to control flare phenomenon. This approach was not used in TITAN study. Therephore in real world all oncologist uses first generation antiandrogen like bicalutamide. We think this is not an useful approach considering the apalutamide mechanism of action. In order to evaluate outcomes we agree that is a limitation of our study, as mentioned in the Discussion at lines 460-463. We updated the manuscript to better clarify this aspect.

Comment 2: [Minor thing: authors refer to Toxicity; ususal it is referred to as Adverse reactions. Toxicity is usually related to severe/extreme adverse reactions with clinical toxicity.]

Response 2: We agree with this comment. Therefore, we replace “toxicity” with “safety” at lines 367, 405, 425 and 431 and with “adverse events” at lines 372, 406, 475. Also, table 2 was modified to replace the word “toxicity” as we agree with the definition of the Reviewer.

Reviewer 2 Report

Comments and Suggestions for Authors

The article "Apalutamide Monotherapy in Metastatic Hormone-Sensitive Prostate Cancer: a Viable Alternative to First Generation Anti-androgen Agents to Avoid the Flare Phenomenon and an Effective Treatment to Achieve Early PSA Response" by Gaetano Facchini et al. lacks a detailed experimental design and results manifestation. The article requires significant/major improvement, particularly in the representation of treatment planning and the dose information for therapeutic agents. The dose accelerated treatment information, and the dose variable treatment output is missing. The PSA level is a method for assessing disease aggressiveness, but it is not a definitive approach. Better diagnostic modalities are well-established to confirm the PC's aggressiveness. Authors need to add their opinions regarding the points mentioned above.

Author Response

Comment 1: [The article "Apalutamide Monotherapy […]” by Gaetano Facchini et al. lacks a detailed experimental design and results manifestation. The article requires significant/major improvement, particularly in the representation of treatment planning and the dose information for therapeutic agents. The dose accelerated treatment information, and the dose variable treatment output is missing.]

Response 1: [Thank you for this comment. We have, accordingly, modified the sections introduction (lines 328-330) and methods (lines 354-355) to emphasize and clarify these points.]

Comment 2: [The PSA level is a method for assessing disease aggressiveness, but it is not a definitive approach. Better diagnostic modalities are well-established to confirm the PC's aggressiveness. Authors need to add their opinions regarding the points mentioned above.]

Response 2: [We agree with this comment. Therefore, we modified introduction at lines 276-280 to emphasize this point. Surely PSA is not completely descriptive of PCa aggressiveness and must be associated with imaging techniques (CT scan, bone scan or next generation imaging); however, in TITAN and SPARTAN trials PSA decline >50% and >90% is a survival surrogate. The standard approach to start treatment in the TITAN study involves bicalutamide for 10–14 days, followed by a Gn-RH analogue plus apalutamide starting on day 15. However, due to the mechanism of action of apalutamide—demonstrating stronger antiandrogen activity compared to bicalutamide—we believe that initiating treatment with bicalutamide monotherapy is not beneficial. It would be preferable to start directly with apalutamide monotherapy and could be a practice change model in real world]

Reviewer 3 Report

Comments and Suggestions for Authors

An early stage observational design clinical study with low patient numbers (with metastatic hormone-sensitive prostate cancer) and carried out in a single cancer treatment centre. With these caveats in mind, the results of the study are interesting and have potential value in future clinical practice if repeated in larger and more comprehensive studies. The limitations of the study are appropriately acknowledged.

The methodology for measurement of PSA and testosterone levels, although a standard procedure, needs to be described in the paper (sample collection and laboratory analysis protocol).

Author Response

Comment 1: [An early stage observational design clinical study with low patient numbers (with metastatic hormone-sensitive prostate cancer) and carried out in a single cancer treatment centre. With these caveats in mind, the results of the study are interesting and have potential value in future clinical practice if repeated in larger and more comprehensive studies. The limitations of the study are appropriately acknowledged.]

Response 1: [Thank you very much for this comment. We agree that further clinical trials with more patients are necessary to examine in depth our results and hopefully change clinical practice.]

Comment 2: [The methodology for measurement of PSA and testosterone levels, although a standard procedure, needs to be described in the paper (sample collection and laboratory analysis protocol).]

 Response 2: [Agree. We added laboratory analysis test at lines 360-361.]

Round 2

Reviewer 1 Report

Comments and Suggestions for Authors

I have no further comments. 

Reviewer 2 Report

Comments and Suggestions for Authors

The updated manuscript may go forward for publication after editorial corrections.